# Inhibitory Effects of Aqueous Ethanol Extracts of Poplar-Type Propolis on Advanced Glycation End Products and Protein Oxidation

**DOI:** 10.3390/foods13193022

**Published:** 2024-09-24

**Authors:** Guangxin Wang, Yu Zhang, Jiangtao Qiao, Hesham R. El-Seedi, Lingjie Kong, Hongcheng Zhang

**Affiliations:** 1State Key Laboratory of Resource Insects, Institute of Apicultural Research, Chinese Academy of Agricultural Sciences, Beijing 100093, China; wangguangxin052@126.com (G.W.); lauralainezy@icloud.com (Y.Z.); jiangtao.qiao@doct.ulige.be (J.Q.); 2Jiangsu Patent Office, Patent Examination Cooperation Center, SIPO, Suzhou 215163, China; 3Jiangsu Beevip Biotechnology Co., Ltd., Taizhou 225300, China; 4Department of Chemistry, Faculty of Science, Islamic University of Madinah, Madinah 42351, Saudi Arabia; hesham@kth.se; 5Key Laboratory of Bee Products for Quality and Safety Control, Ministry of Agriculture and Rural Affairs, Beijing 100093, China

**Keywords:** protein glycation, poplar-type propolis, flavonoids, diabetic complications

## Abstract

(1) Background: The non-enzymatic glycation of proteins is a significant contributor to the formation of advanced glycation end products (AGEs) and intermediates that are responsible for diabetic complications. It is imperative to explore effective inhibitors to prevent protein glycation. (2) Methods: This study aimed to investigate the inhibitory potential of various aqueous ethanol extracts of poplar-type propolis on AGEs and oxidative modifications in bovine serum albumin (BSA)-glucose and BSA-methylglyoxal models. (3) Results: The results revealed that these propolis extracts exhibited significant effectiveness in inhibiting the formation of total AGEs, pentosidine, and Nε-carboxymethyllysine (CML). Furthermore, the investigation discovered that these propolis extracts can effectively inhibit oxidative modification, based on measuring the levels of carbonyl and thiol groups and analyzing tryptophan fluorescence quenching. Notably, 75% ethanol extracts of propolis (EEP) exhibited the highest inhibitory activity, surpassing the chemical inhibitor aminoguanidine (AG). (4) Conclusions: The remarkable anti-glycation potency of aqueous ethanol extracts of poplar-type propolis can be attributed to their elevated contents of phenolic compounds, especially abundant flavonoids, which inhibit the formation of AGEs by scavenging free radicals, decreasing the levels of reactive oxygen species (ROS), and capturing reactive carbonyl species (RCS) in the protein glycation process. Our results indicate that poplar-type propolis may be a potential AGE inhibitor and could be used to develop functional foods and nutraceuticals to prevent diabetic complications.

## 1. Introduction

Diabetes represents the most prevalent endocrine disorder, characterized by abnormal elevations in plasma glucose levels. Forecasts indicate a substantial upsurge in diabetes prevalence in the forthcoming years, primarily attributable to progressively prosperous lifestyles, notably observed in developing countries. Prior studies have documented that the global population afflicted with diabetes will escalate to 366 million by 2030, with diabetes-related fatalities accounting for approximately 9% of the global mortality rate [1]. Diabetes gives rise to a series of complications, such as retinopathy, cataracts, atherosclerosis, neuropathy, nephropathy, and impaired wound healing. These complications stem primarily from the accumulation of advanced glycation end products (AGEs) within human tissues [2]. AGEs can be generated through a sequence of condensation reactions subsequent to protein glycation. Protein glycation, a non-enzymatic process, occurs when the carbonyl groups of reducing sugars react with the free amino groups of proteins [3]. This intricate process involves the interactions between aldehyde groups found in reducing sugars, such as glucose and fructose, and amino groups present in proteins, enzymes, nucleic acids, and phospholipids, leading to the formation of a reversible Schiff base. Subsequently, these adducts undergo the Amadori rearrangement, transforming into stable Amadori products. The Amadori products undergo a further series of reactions involving dicarbonyl intermediates to form AGEs [4]. The accumulation of AGEs within human tissues and organs is associated with oxidative stress and inflammation toxic to organisms. Notably, hemoglobin glycation can result in modifications to its secondary structure and even the degradation of heme groups. Moreover, these structural alterations can impair the functional efficacy of hemoglobin and other proteins, leading to various complications in vivo, such as aging, cataracts, uremia, and arthritis [5]. Therefore, inhibiting protein glycation appears to be very crucial in preventing these complications.

Despite the extensive testing of synthetic drugs and chemical agents for glycation inhibition, their clinical application has been limited due to the prevalence of numerous side effects and low efficacy. A specific example is aminoguanidine (AG), an artificial nucleophilic hydrazine compound, which has been validated to inhibit the formation of AGEs and prevent the progression of diabetic complications. However, previous studies have indicated that AG can cause serious adverse effects in diabetic individuals, such as congestive heart failure, anemia, and gastrointestinal disturbance [6]. Thus, the exploration of natural AGE inhibitors especially from herbs and dietary plants becomes imperative.

Propolis, as a resinous substance collected from diverse plant species and mixed with honey bee salivary secretions, has been used in folk medicine since ancient times. Propolis confers various health benefits, including antimicrobial, antifungal, anti-aging, anticancer, anti-inflammatory, and antioxidant activities [7]. Recent research has also highlighted the anti-glycation effects of propolis [8,9]. These functional properties can be attributed to the presence of over 300 natural bioactive compounds, mainly including flavonoids, phenolic aldehydes, and terpenoids. However, the bioactive compounds in propolis vary significantly depending on geographical and botanical origins. For example, Brazilian green propolis is predominantly composed of artepillin C and terpenoids [10]; red propolis in Mexico is composed of flavanones, isoflavans, and pterocarpans [11]; aspen propolis from northern regions of Europe is composed of flavonoids and phenolic acids [12]. In recent decades, China has emerged as a prominent global producer of propolis. Multiple studies have indicated that the Chinese propolis belongs to the poplar-type; in other words, the primary plant origin of Chinese propolis is the Populus species [13]. However, the precise mechanism by which poplar-type propolis inhibits glycation remains uncertain.

Previous investigations have explored the phenolic composition, botanical origins [14], antioxidant activities [15,16], and anti-α-glucosidases properties of extracts from poplar-type propolis [17]. The objective of this present study was to assess the potential of different aqueous ethanol extracts of poplar-type propolis in inhibiting glycation, encompassing their effects on AGE formation and protein oxidative modification.

## 2. Materials and Methods

### 2.1. Chemicals and Materials

D-glucose, trichloroacetic acid, dimethyl sulfoxide (DMSO), glyoxylic acid, NaN_3_, sodium cyanoborohydride (NaCNBH_3_), methylglyoxal (MGO), nitro blue tetrazolium (NBT), 2,4-dinitrophenylhydrazine (DNPH), L-cysteine, 5,5′-dithiobis(2-nitrobenzoic acid) (DNTB), aminoguanidine (AG), bovine serum albumin (BSA), trichloroacetic acid (TCA) were purchased from Sigma Chemical Co. (St. Louis, MO, USA). Nε-(carboxymethyl)lysine (CML) ELISA Kit was purchased from CycLex Co., Ltd. (Tokyo, Japan). Methanol and acetic acid of HPLC grade were purchased from Thermo Fisher Scientific Inc. (Fair Lawn, NJ, USA). Ultrapure water was purified using a Milli-Q-Integral System (Millipore, Billerica, MA, USA). Ethyl alcohol of analytical grade and all other reagents were from Solarbio Inc. (Beijing, China).

3,4-Dihydroxybenzaldehyde (1), caffeic acid (2), vanillin (3), *p*-coumaric acid (4), ferulic acid (5), isoferulic acid (6), benzoic acid (7), 3,4-dimethoxy cinnamic acid (8), cinnamic acid (9), 4-methoxy cinnamic acid (10), pinobanksin (12), quercetin (13), alpinetin (14), kaempferol (15), apigenin (17), isorhamnetin (18), pinocembrin (19), chrysin (22), phenethyl caffeate (23), galangin (24) and cinnamyl cinnamate (31) were purchased from Sigma-Aldrich Chemical Co. (St. Louis, MO, USA). Cinnamylidene acetic acid (16) was purchased from Funakoshi Chemical Co. (Tokyo, Japan). 9-oxo-10(E),12(Z)-octadecadienoic acid (9-oxo-OCLE) (33) and 9-oxo-10(E),12(E)-octadecadienoic acid (9-oxo-OCLA) (34) were obtained from Cayman Chemical Co. (Ann Arbor, MI, USA). Pinobanksin-3-O-acetate (21), pinostrobin (28), and tectochrysin (29) were purchased from BioBioPha Co., Ltd. (Kunming, China). 5-methoxy pinobanksin (11), benzyl caffeate (20), benzyl-*p*-coumarate (25), benzyl ferulate (26), cinnamyl caffeate (27), Cinnamyl-*p*-coumarate (30), 4-methoxycinnamyl cinnamate (32) were collected by preparative HPLC.

### 2.2. Propolis Samples

The crude propolis samples were collected in hives of *Apis mellifera* from the apiary located inside China National Botanical Garden (Beijing, China) in August 2021, and poplar trees around the apiary were identified as *Populus canadensis* Moench by the plant taxonomist of China National Botanical Garden [14]. After collection, the propolis samples were stored in a refrigerator at −18 °C until analysis.

### 2.3. Preparation of Propolis Aqueous Ethanol Extracts

Propolis extracts were prepared according to the literature with slight modifications [16]. The frozen propolis samples were crushed into homogeneous powder using a pulverizer (XFB-500, Zhongzhou Co., Chongqing, China). Then, the powdered propolis sample (10 g) was extracted for 5 h under ultrasonication (DCTZ-1000, Hongxianglong Biotechnology Co., Ltd., Beijing, China) at a frequency of 60 kHz and power of 100 W using different solvents (200 mL), including water, 25, 50, 75, 95, and 100 wt.% ethanol/water solvents. The suspensions were centrifuged at 400× *g* for 5 min to obtain supernatants. Then the supernatants were concentrated using a rotary evaporator (Rotavapor, R-215, Buchi Co., Ltd., Flawil, Switzerland) and finally freeze-dried as dried extracts. These dried extracts were, respectively, expressed as WEP for water extracts of propolis; 25% EEP, 50% EEP, 75% EEP, 95% EEP, and 100% EEP for 25, 50, 75, 95, and 100 wt.% ethanol/water extracts. Before the anti-glycation experiments, these dried extracts were re-dissolved in a small amount of DMSO, and further diluted with phosphate buffer to the appropriate concentration.

### 2.4. Phenolics Determination of Propolis Extracts Using HPLC-PDA

Different dried extracts of 0.5 g (WEP, 25% EEP, 50% EEP, 75% EEP, 95% EEP, and 100% EEP) were re-dissolved using methanol (10 mL), then filtered using MILLEX-GA 0.22 μm filter. The profiles of propolis extracts were analyzed by HPLC (PDA-20A diode array detector, SIL auto-injection valve, CTO-10A thermostat, and pump LC-6AD, Shimadzu, Tokyo, Japan) at a wavelength of 280 nm. The 10 μL solution was loaded and eluted through a C18 reversed-phase column (150 mm × 4.6 mm 3 μm Gemini, Phenomenex, Inc., Torrance, CA, USA) with a gradient elution. The mobile phase consisted of water (phase A) and methanol (phase B), containing 2% acetic acid. The temperature of the column oven was set at 35 °C. A 150 min linear gradient with a flow rate of 0.65 mL/min was programmed as follows: 0–10 min, 22–32% B; 10–25 min, 32–35% B; 25–35 min, 35–38% B; 35–52 min, 38–51% B; 52–70 min, 51–52% B; 70–80 min, 52–52% B; 80–90 min, 52–53% B; 90–100 min, 53–59% B; 100–115 min, 59–63% B; 115–130 min, 63–75% B; 130–150 min, 75–80% B. The compounds were identified by comparing their retention time and UV spectra with commercial standards [14,15,18]. The compounds were quantified using external calibration curves with pure standards, based on the peak areas at a wavelength of 280 nm.

### 2.5. Effect of Propolis Extracts on Hydroxyl Radical Produced by Glucose Autoxidation

The effect of the propolis extracts on hydroxyl radical produced by glucose autoxidation was assayed according to the procedure described in the literature [19]. Briefly, the experiment was conducted by adding different dried extracts at a range of concentrations (0, 6.25, 12.5, 25, 50, 100 μg/mL) to a 4 mL reaction mixture consisting of 1 mM sodium benzoate, 100 mM potassium phosphate buffer (pH 7.2), 500 mM glucose, and 0.1 mM CuSO_4_. The mixture was then incubated at 37 °C for four days. The decrease in benzoate hydroxylation, measured by the fluorescence intensity (excitation and emission maxima of 308 nm and 410 nm, respectively), correlates with the hydroxyl radical scavenging activity of the propolis extracts. Control measurement of the reaction mixture consisted of glucose. The inhibition rate was calculated using the following equation, and half inhibition concentration (IC_50_) values were used to express the inhibitory effect:inhibition rate (%) = [1 − (Asample/Acontrol)] × 100% (1)

### 2.6. Anti-Glycation Determination of Propolis Extracts in BSA Glycated Models

#### 2.6.1. BSA-Glucose/BSA-MGO Models

The BSA and glucose model (BSA-glucose) was established based on a previously reported method [20]. In brief, a total of 50 mg/mL BSA, 0.8 M glucose, and 0.2 g/L NaN_3_ were dissolved in phosphate buffer (0.2 M, pH 7.4). Different dried extracts at concentrations of 0 and 2.5 μg/mL were added into protein-glucose solutions, and then incubated at 37 °C for 7 days. After incubation, the inhibitory effects on total AGEs, pentosidine, and Nε-(carboxymethyl)lysine (CML) were measured. AG (1 mM) was used as the positive control. NaN_3_ was added to the protein-glucose solutions to prevent bacterial contamination.

The BSA and MGO model (BSA-MGO) was established using the method reported in [21]. A total of 1 mg/mL BSA, 5 mM MGO, and 0.2 g/L NaN_3_ were dissolved in phosphate buffer (0.1 M, pH 7.4). Different dried extracts at concentrations of 0 and 2.5 μg/mL were added into protein-MGO solutions and then incubated at 37 °C. After 7 days, the inhibitory effect on total AGEs was measured. In addition, AG (1 mM) was also used as the positive control.

#### 2.6.2. Measurement of Total AGEs, Pentosidine and CML

Total AGEs and pentosidine in the protein reaction solutions were analyzed by determining the fluorescence intensity at λex/λem = 330/410 nm and λex/λem = 335/385 nm, respectively, with a Hitachi F-4600 fluorescent spectrometer (Hitachi Corporation, Tokyo, Japan). The inhibition rate of total AGEs or pentosidine was calculated by the equation:inhibition rate (%) = [1 − (fluorescence of the solution with inhibitors/fluorescence of the solution without inhibitors)] × 100% (2)

The measurement of CML in the protein reaction solutions was conducted according to the literature [22]. The BSA incubation with glucose and different dried extracts was carried out as described in Section 2.6.1. Then, the protein solution was combined with 45 mM glyoxylic acid and 150 mM NaCNBH_3_ prepared in phosphate buffer (0.1 M, pH 7.4), resulting in a final volume of 4 mL. The solution was incubated for 24 h at 37 °C. The level of CML was measured using CML ELISA Kit (CycLex Co., Ltd., Tokyo, Japan). The operation of the ELISA Kit was performed according to the instructions, and the inhibition rate of CML was calculated by the following equation:inhibition rate (%) = [1 − (absorbance of the solution with inhibitors/absorbance of the solution without inhibitors)] × 100% (3)

### 2.7. Determination of Fructosamine

Fructosamine can serve as an indicator of the early glycation of BSA induced by glucose. The fructosamine level in the BSA-glucose system was measured using an NBT assay method [23]. In accordance with Section 2.6.1, the BSA was incubated with glucose and different dried extracts. Then, the protein solution was mixed with 1 mL NBT reagent (0.5 mM) dissolved in sodium carbonate buffer (0.2 M, pH 10.4) and incubated for 1 h at 37 °C. Absorbance at 530 nm was immediately measured using a UV-2500/ultraviolet–visible spectrophotometer (Shimadzu Co., Ltd., Tokyo, Japan). The inhibition rate of fructosamine was calculated by the following equation:inhibition rate (%) = [1 − (absorbance of the solution with inhibitors/absorbance of the solution without inhibitors)] × 100%(4)

### 2.8. Determination of Protein Carbonyl and Thiol Group Levels

The inhibitory effect of propolis extracts on protein oxidative modification was evaluated by detecting protein carbonyl and thiol group levels of BSA oxidative modifications in the glycoxidation process following the method described in the literature with slight modifications [24]. In accordance with Section 2.6.1, the BSA was incubated with glucose and different dried extracts. Then, the protein solution was mixed with 1 mL of 10 mM 2,4-dinitrophenyl -hydrazine (DNPH) prepared in 2 M HCl, and incubated for 30 min at room temperature. After incubation, 1 mL of cold TCA (10%, *w*/*v*) was added to the mixture and centrifuged at 3000× *g* for 10 min. The protein pellet was washed three times with 2 mL of ethanol/ethyl acetate (1:1, *v*/*v*) and dissolved in 1 mL of guanidine hydrochloride (6 M, pH 2.3). The concentration of carbonyl was calculated based on the molar extinction coefficient of DNPH (ε370 = 22,000 cm^−1^M^−1^) using a UV-2500/ultraviolet–visible spectrophotometer. The data are expressed as nmol of carbonyl per mg protein.

Thiol group levels were estimated based on the thiol/disulfide reaction of thiol and Ellman’s reagent, 5,5′-dithiobis (2-nitrobenzoic acid) (DNTB). After incubation of BSA with glucose and different dried extracts, the protein solution was mixed with 2.5 mM DTNB prepared in phosphate buffer (0.2 M, pH 7.4) and incubated for 15 min at room temperature. Absorbance at 410 nm was measured using a UV-2500/ultraviolet–visible spectrophotometer. The standard curve was prepared by using various concentrations of L-cysteine, and the concentration of the thiol group was calculated based on the standard curve. The data are expressed as pmol of thiol group per mg protein.

### 2.9. Determination of Protein Conformation Changes

The tryptophan fluorescence quenching assay was used to evaluate protein conformation change during the glycoxidation process, according to a procedure described previously [21]. The BSA-MGO system was prepared following Section 2.6.1, with the addition of different dried extracts at concentrations of 1.25 and 2.5 μg/mL, respectively. The tryptophan fluorescence quenching by glycation was determined as fluorescence spectra at Ex 280 nm using a Hitachi F-4600 fluorescent spectrometer. In addition, three-dimensional fluorescence spectra were also measured under the following conditions: the emission wavelength was recorded between 200 and 700 nm, the initial excitation wavelength was set to 200 nm with increment of 50 nm for each scanning curve.

### 2.10. Statistical Analysis

For statistical analysis, all experiments were performed in triplicate. The values are presented as means ± standard deviation and significance differences at *p* < 0.05 among treatment means were obtained with SPSS software (version 16.0, SPSS GmbH Software, Munich, Germany).

## 3. Results and Discussion

### 3.1. The Quantification of Phenolic Compounds in Propolis Extracts

Propolis contains abundant phenolic compounds, which confer upon it a wide range of functional activities, such as antioxidant, anti-inflammatory, antimicrobial, antifungal, and anti-diabetic properties. Consequently, the identification of phenolic compounds in propolis is crucial for the comprehensive exploration of its potential properties. In this investigation, the phenolic composition of aqueous ethanol extracts of poplar-type propolis was analyzed utilizing HPLC-PDA. Figure 1 displays the HPLC chromatograms of propolis extracts obtained using different ethanol/water solvents, revealing notable variations. These chromatograms can be categorized into two distinct types. The first type, comprising WEP (Figure 1B) and 25% EEP (Figure 1C), exhibited a limited number of peaks mainly occurring prior to a retention time of 50 min. The second type encompasses 50% EEP (Figure 1D), 75% EEP (Figure 1A), 95% EEP (Figure 1E), and 100% EEP (Figure 1F), which exhibit a greater abundance of peaks within the retention time range of 50 to 140 min. The findings imply that ethanol concentrations of 50% or higher appear to facilitate the extraction of a broader range of components present in propolis, including flavonoids, phenolic acids, and their esters; in contrast, only phenolic acids present in WEP and 25% EEP, such as caffeic acid, *p*-coumaric acid, ferulic acid, isoferulic acid, and 3,4-dimethoxy cinnamic acid. The extraction efficiency of 75 wt.% ethanol/water was found to be the highest, as evidenced by the presence of the highest number of compound species and contents (Figure 1A). Specifically, 34 compounds, comprising 19 phenolic acids and their esters, 13 flavonoids, and 2 unsaturated hydrocarbons, were identified in the 75% EEP (Figure 1 and Table 1). Flavonoids were observed to be the predominant components in the 75% EEP, especially with pinobanksin-3-acetate, pinocembrin, chrysin, galangin, 5-methoxy pinobanksin, and pinobanksin. Among these, pinobanksin-3-O-acetate exhibited the highest content of 196.91 mg/g. The sum of the quantified 13 flavonoids exceeded 50% of the total weight. Previous literature has reported that flavonoid compounds such as chrysin, galangin, pinocembrin, pinobanksin and its ester derivatives exhibit strong antioxidant activities and are associated with reduced risk of various diseases, including heart disease, asthma, cancer, diabetes, and Alzheimer’s [25]. For instance, pinobanksin-3-cinnamate may provide neuroprotection through counteracting oxidative stress and has the potential to treat vascular dementia [26]. Pinocembrin has been shown to inhibit atherosclerosis progression by enhancing the level and function of endothelial progenitor cells [27]. Our previous research demonstrated that galangin and pinocembrin from propolis can improve insulin resistance in HepG2 cells [28]. Additionally, some flavonoids, such as quercetin and genistein, have been confirmed to inhibit AGEs [29]. Therefore, we speculated that the high content of flavonoids in the poplar-type propolis, including pinocembrin, chrysin, galangin and pinobanksin and its ester derivatives, etc., may be the primary trigger for its inhibitory effects against protein glycation.

### 3.2. Effect of Propolis Extracts on Hydroxyl Radicals from Glucose Autoxidation

It is widely acknowledged that the presence of reactive oxygen species (ROS) and free radicals can contribute to the development of various chronic human ailments, including aging, heart disease, and cancer [30]. Previous studies have reported that elevated levels of glucose in bodily fluids, such as blood, can lead to oxidative harm, resulting in an imbalance between ROS production and the antioxidant defense mechanisms within biological systems. The oxidation process and the generation of free radicals have been observed to occur during protein glycation [31]. Therefore, the inhibition of free radical generation derived from the glycation process and the subsequent inhibition of the protein oxidative modification are considered to be the important mechanisms of the anti-glycation. The present study sought to assess the efficacy of various aqueous ethanol extracts of poplar-type propolis in scavenging hydroxyl radicals generated through glucose autoxidation. The IC_50_ values, as shown in Table 2, ranged from 9.99 to 17.44 μg/mL for the different propolis extracts. It is worth mentioning that among the various aqueous ethanol extracts, 75% EEP exhibited the greatest capacity for scavenging hydroxyl radicals with an IC_50_ value of 9.99 μg/mL, whereas WEP showed the weakest ability. Previous studies have established a correlation between ethanol/water concentrations and the quantity and composition of phenolic compounds in food or plant extracts [32]. Our results in Figure 1 showed that the higher concentrations of ethanol (≥50%) appear to facilitate the extraction of a wider range of phenolic compounds, including flavonoids, phenolic acids, and their esters, from propolis. Conversely, only phenolic acids were extracted in WEP and 25% EEP. Furthermore, the extracts of poplar-type propolis exhibited a significant presence of flavonoids, accounting for approximately 50% of the total weight (Table 1). This viewpoint is further supported by the content sums of 13 flavonoids, 19 phenolic acids and their esters, shown in Table 2. Thus, the scavenging capacity of hydroxyl radicals for these propolis extracts appears to be associated with their rich flavonoid content. These findings are consistent with previous reports which have shown a positive correlation between the concentration of flavonoids and the scavenging free radicals derived from the glycoxidation process [31].

### 3.3. Effect of Propolis Extracts on Total AGEs

Multiple mechanisms have been identified as contributors to hyperglycemia-induced oxidative stress, including glucose autoxidation, protein oxidative modification, and the formation of AGEs [33]. The accumulation of AGEs in vivo can trigger cellular apoptosis, tissue injury, and organ damage, particularly in individuals with diabetes mellitus. When proteins are exposed to reducing sugars (glucose, ribose), a natural and non-enzymatic glycation process occurs to form AGEs. Additionally, reactive carbonyl species (RCS), such as 3-deoxyglucosone, glyoxal, and methylglyoxal (MGO), play a crucial role as intermediates during the protein glycation process. The RCS can readily bind to amino groups, resulting in the modification of biological molecules to form covalently cross-linked aggregates [34]. Similar to the glycation of proteins by glucose, the glycation of amino acids by RCS is also an essential pathway for the formation of AGEs. In this study, the BSA-glucose and BSA-MGO models were selected for evaluating the anti-AGE effects of aqueous ethanol extracts of poplar-type propolis. As shown in Figure 2, different propolis extracts demonstrated varying degrees of inhibition on total AGEs in both models. The inhibitory rates followed the order of 75% EEP > 50% EEP > 95% EEP > 100% EEP > 25% EEP > WEP. The 75% EEP exhibited the highest inhibitory effect in preventing BSA glycation by glucose-MGO (74.05% inhibition rate at 2.5 μg/mL); in comparison with maize extracts (50% inhibition rate at 9.5 μg/mL) [35]. These findings suggest that the remarkable inhibitory glycation effect of the propolis extracts may be attributed to the presence of abundant phenolics, particularly flavonoids, which are present in high concentrations (≥50%) in the 75% EEP. The outstanding inhibitory effect of these propolis extracts may be related to abundant phenolics, especially flavonoids in high concentrations (≥50%) of EEP (Table 2). Previous studies have indicated that plant extracts containing large amounts of phenolic compounds can effectively hinder the formation of AGEs [25]. Our results in Table 2 demonstrate that these propolis extracts can effectively scavenge hydroxyl radicals generated through glucose autoxidation. Of note, the inhibitory effects on the formation of total AGEs exhibited by 50% EEP, 75% EEP, 95% EEP, and 100% EEP were significantly superior to those of the positive control AG (*p* < 0.05). As an antioxidant and nucleophilic agent, AG possesses the potential to trap RCS. Thus, the inhibition by poplar-type propolis extracts on the formation of AGEs may involve scavenging radicals and trapping RCS.

### 3.4. Effect of Propolis Extracts on Pentosidine and CML

AGEs, which consist of a diverse range of chemical structures, encompass both fluorescence and non-fluorescence compounds, including argpyrimidine, crossline, pyrropyridine, pentosidine, and CML. Pentosidine and CML specifically serve as representative examples of fluorescence and non-fluorescence compounds, commonly employed in AEG research. In this study, the inhibitory effects of different aqueous ethanol extracts of poplar-type propolis on the formation of pentosidine and CML were evaluated using the BSA-glucose system. As shown in Figure 3, the inhibitory rates of pentosidine vary across different propolis extracts, ranging from 47.46% to 76.02% (Figure 3A). The highest inhibitory rate for pentosidine was observed in the 75% EEP extract (76.02%). Furthermore, the inhibitory rates of high concentrations (≥50%) of EEP were higher than that of the AG solution (63.55%), except for the 25% EEP and WEP extracts. Conversely, all aqueous ethanol extracts of propolis exhibited similar CML inhibition, with inhibitory rates ranging from 32.19% to 35.3%, compared to AG’s of 33.72%, and no statistically significant differences were observed.

Elevated levels of pentosidine and CML are associated with diabetic complications, such as diabetic retinopathy, cardiovascular disease, kidney disease, and neuropathy in diabetic patients. Therefore, diabetic patients usually monitor pentosidine and CML levels and take steps to diminish them [36]. Our results indicated that higher concentrations (≥50%) of EEP exhibited a more pronounced inhibitory effect on pentosidine compared to lower concentrations (≤25%) of EEP. This enhanced inhibitory activity can be attributed to the higher levels of flavonoid compounds in the propolis extracts. Additionally, our results also revealed that these propolis extracts have inhibitory effects against CML similar to AG, and they exhibit higher inhibitory activity against pentosidine compared to CML. Research has provided evidence indicating that pentosidine is predominantly synthesized via a sequence of oxidative reactions following the amalgamation of RCS with lysine and arginine residues. Furthermore, apart from its formation through protein oxidation, CML can also be generated through various pathways [37]. The propolis extracts might inhibit CML formation by trapping a certain amount of dicarbonyl compounds and inhibiting the subsequent series of oxidation and cross-linking reactions between proteins and dicarbonyl compounds, leading to a relatively weaker inhibitory effect [38]. The findings suggest that the presence of phenolic compounds in poplar-type propolis enables the sequestration of RCS, diminishing the formation of two prevalent AGEs: pentosidine and CML. Consequently, poplar-type propolis seems to emerge as a natural AGE inhibitor for diabetic patients.

### 3.5. Effect of Propolis Extracts on Fructosamine

Amadori products, which are intermediate compounds formed during the initial stage of glycation, possess the capability to reduce NBT (nitro blue tetrazolium) and generate a tetrazolyl radical. As Amadori products, fructosamine is frequently employed as an indicator to assess the inhibitory activity on the early-stage glycation reaction. Figure 3C presents the inhibitory effect of various aqueous ethanol extracts of propolis on fructosamine formation. The results showed that the inhibitory rates of different propolis extracts were either comparable to or slightly higher than that of 1 mM AG solution. Particularly, 100% EEP exhibited the highest inhibitory activity with the lower inhibitory rate 26.33%.

The glycation reactions can be categorized into two distinct phases: the initial and final phase. During the initial stage of the reaction, Amadori products are synthesized without the generation of ROS and RCS. In the subsequent stage, these Amadori products can undergo oxidative processes, fragmentation, and chemical rearrangement to form various dicarbonyl compounds, finally leading to the formation of AGEs [3]. Oxidation reactions play a crucial role during this phase. Our results demonstrated that propolis extracts exhibited a modest inhibitory effect on the formation of Amadori products. Notably, 100% EEP displayed the highest inhibitory activity. Previous studies indicated that due to differences in the molecular structures of phenolic acids and flavonoids, their mechanisms for inhibiting AGEs may differ [25]. Of note, phenolic esters have also been reported to exhibit an inhibitory effect on AGEs [39]. We speculated that the relatively higher inhibitory activity of 100% EEP may be attributed to its elevated content of phenolic acids and esters (Table 2). On the other hand, the results in Figure 2 and Figure 3 showed that these propolis extracts can effectively impede the production of total AGEs, pentosidine and CML. Consequently, our findings suggest that the anti-AGE impact of poplar-type propolis extracts predominantly stems from their ability to suppress the oxidative reactions during the later stages of glycation.

### 3.6. Effect of Propolis Extracts on Protein Oxidative Modification

Maintaining the redox equilibrium of proteins is of utmost importance to guarantee their optimal functionality within cellular processes. The process of glycation, in which proteins undergo modification by free radicals and glucose, can disrupt the functioning of essential biomolecules, such as enzymes, receptors, and membrane transporters [40]. During the later stages of glycation, proteins can undergo oxidative modifications due to the presence of ROS generated either by the autoxidation of glucose or the oxidative degradation of Amadori intermediates [41]. These oxidative modifications can lead to the formation of protein carbonyls and the loss of thiol groups, further destroying the protein’s functionality. This study measured the levels of carbonyl and thiol groups in the oxidative modifications of BSA during the glycoxidation process. Table 3 demonstrates a notable increase in protein carbonyls and a decrease in thiol groups, indicating the occurrence of oxidative modifications in BSA when incubated with glucose. However, when adding different extracts of propolis to this reaction system, the levels of carbonyl significantly reduced. Importantly, all propolis extracts exhibited superior inhibitory effects compared to the positive control AG. Among the various extracts, the 75% EEP exhibited the highest potency, achieving an inhibitory rate of 36.12%. Additionally, the 75% EEP also exhibited the strongest ability to inhibit thiol group oxidation, with an inhibitory rate of 78.01%, surpassing the positive control AG. The results suggest that poplar-type propolis extracts possess the capacity to suppress protein oxidative modification during the glycoxidation process. The decrease in the levels of ROS can be attributed to the strong antioxidant properties of the phenolic compounds in these propolis extracts, showing further responsibility for their inhibitory effect on AGE formation.

### 3.7. Effect of Propolis Extracts on Protein Conformation Changes

The tryptophan fluorescence quenching method is a commonly employed technique for evaluating protein conformational changes caused by glycation. This study aimed to investigate the impact of poplar-type propolis extracts on the quenching of tryptophan fluorescence spectra in BSA protein induced by MGO. Figure 4 shows that the introduction of MGO to the reaction system resulted in a substantial reduction in BSA’s tryptophan fluorescence, with a quenching rate exceeding 90%, accompanied by a red shift in the fluorescence peaks. This indicates that the interaction between RCS and BSA induced the alterations of protein conformation. However, the quenching of tryptophan fluorescence induced by MGO was observed to decrease significantly in a concentration-dependent manner, when treated with various aqueous ethanol extracts of propolis. Noteworthily, 75% EEP exhibited the most pronounced inhibitory effect among all propolis extracts, comparable to the positive control AG. Additionally, three-dimensional fluorescence spectra serve as spatial maps utilizing excitation wavelength, emission wavelength, and fluorescence intensity as corresponding coordinates. Compared to conventional fluorescence spectra, three-dimensional fluorescence spectra offer more detailed information about the changes of protein conformation. Figure 5 illustrates the three-dimensional fluorescence spectra and contour maps of the BSA protein induced by MGO after adding propolis extracts. The contour map displays a comprehensive overview of the fluorescence spectra. Distinct fluorescence peaks of BSA can be easily observed in the three-dimensional fluorescence spectra and contour maps (Figure 5A). However, upon incubation with MGO, significant fluorescence intensity reduction and peak position shifts can be observed, accompanied by a transition of λex and λem from 280 nm and 334 nm to 340 nm and 410 nm (Figure 5B). These changes indicate that the conformations of the peptide backbone, tryptophan and tyrosine residues of BSA were modified [42]. When various propolis extracts were introduced into the BSA-MGO system, the overall changes in λex and λem were not significant, but variations were observed between the different propolis extracts (Figure 5C–H). Notably, the three-dimensional fluorescence spectra for 75% EEP exhibited the highest resemblance to that of AG, consistent with the results derived from the tryptophan fluorescence quenching analysis (Figure 4). The findings in Figure 4 and Figure 5 further confirm that the extracts of poplar-type propolis possess a discernible ability to trap RCS, consequently mitigating the formation of AGEs during the late-stage protein glycation. Furthermore, it is worth noting that the flavonoids present in these propolis extracts may exert a more pronounced influence on RCS capture in comparison to phenolic acids and their esters.

Overall, in this study, we investigated the anti-glycation effect and mechanism of aqueous ethanol extracts of poplar-type propolis. Our findings indicate that these extracts can effectively scavenge hydroxyl radicals generated through glucose autoxidation. Additionally, they also exhibit a significant inhibitory effect on total AGEs, pentosidine, and CML formation in the glycation model. However, their inhibitory activity towards Amadori products was comparatively weak. Furthermore, based on carbonyl and thiol group changes and tryptophan fluorescence quenching analysis, we found that these propolis extracts could effectively suppress protein oxidative modification and maintain protein conformation during the glycation process. Among the various aqueous ethanol extracts, 75% EEP exhibited the best inhibitory activity.

AGE inhibitors can prevent glycation through various mechanisms, including the prevention of Amadori product formation, inhibition of the late phase of glycation reaction, or interference with the attachment of sugars to proteins. Furthermore, their ability to scavenge free radicals and to break cross-links may be responsible for inhibiting glycation [43]. Our findings suggest that the inhibitory effects on AGEs and protein oxidative modification observed in aqueous ethanol extracts of poplar-type propolis may be attributed to the abundant phenolics, which can decrease the formation of AGEs by the scavenging of free radicals, decreasing the levels of ROS, and effectively capturing RCS in the protein glycation process [44]. Furthermore, our findings also demonstrated a significant correlation between the anti-glycation properties of various aqueous ethanol extracts and the concentration of flavonoids. This observation aligns with a previous study suggesting that the addition of flavonoids during the glycoxidative process effectively inhibits protein glycation and oxidation. Thus, abundant flavonoids seem to endow propolis extracts with the ability to possess an inhibitory impact on glycation [31].

Previous research has established a connection between elevated levels of AGEs in human tissue and the development of various diseases such as obesity, diabetes, inflammation, and cardiovascular conditions. This underscores the importance of AGE inhibition for managing these health issues. Due to the significant adverse effects of synthetic AGE inhibitors, there has been a growing interest in discovering and developing natural alternatives in recent years. Our current work suggests that the natural product propolis, especially poplar-type propolis, may offer a viable strategy for AGE inhibition. Given its specific phenolic compounds, poplar-type propolis has the potential to be a promising candidate as an AGE inhibitor in dietary supplements and nutraceuticals. In the future, we will focus on clinical trials to evaluate the efficacy of poplar-type propolis in inhibiting AGEs in vivo, providing reliable evidence for the development of novel propolis nutraceuticals aimed at preventing and treating chronic diseases associated with AGEs.

## 4. Conclusions

Our findings reveal that the aqueous ethanol extracts of poplar-type propolis exhibit significant anti-glycation activity. In particular, 75% EEP exhibited the most pronounced inhibitory effect on the formation of AGEs. This potent inhibitory effect can be attributed to the high concentration of flavonoids present in 75% EEP, which contribute to its antioxidant properties and ability to trap RCS. These findings provide reliable evidence supporting the potential use of poplar-type propolis as a promising agent for inhibiting AGEs formation, thereby offering a viable avenue for the development of novel nutraceuticals aimed at preventing diabetic complications.

## Figures and Tables

**Figure 1 foods-13-03022-f001:**
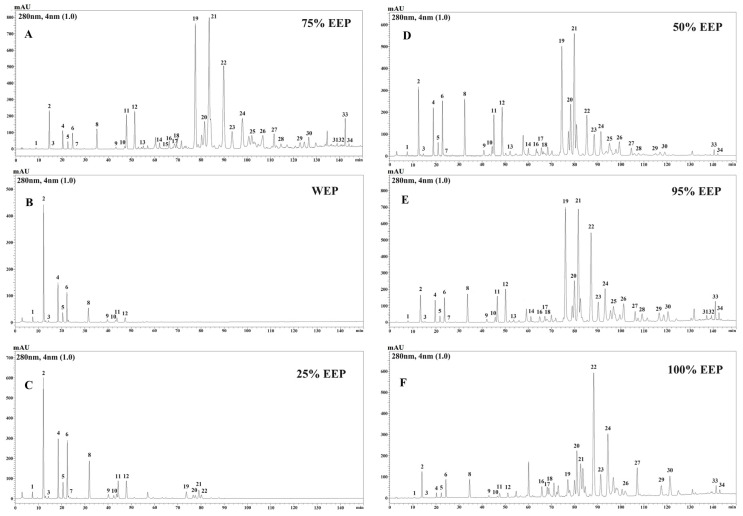
HPLC chromatograms of different aqueous ethanol extracts of propolis: 75% EEP (**A**), WEP (**B**), 25% EEP (**C**), 50% EEP (**D**), 95% EEP (**E**), 100% EEP (**F**). (1) 3,4-Dihydroxybenzaldehyde; (2) caffeic acid; (3) vanillin; (4) p-coumaric acid; (5) ferulic acid; (6) isoferulic acid; (7) benzoic acid; (8) 3,4-dimethoxy cinnamic acid; (9) cinnamic acid; (10) 4-methoxy cinnamic acid; (11) 5-methoxy pinobanksin; (12) pinobanksin; (13) quercetin; (14) alpinetin; (15) kaempferol; (16) cinnamylidene acetic acid; (17) apigenin; (18) isorhamnetin; (19) pinocembrin; (20) benzyl caffeate; (21) pinobanksin-3-O-acetate; (22) chrysin; (23) phenethyl caffeate; (24) galangin; (25) benzyl p-coumarate; (26) benzyl ferulate; (27) cinnamyl caffeate; (28) pinostrobin; (29) tectochrysin; (30) cinnamyl-p-coumarate; (31) cinnamyl cinnamate; (32) 4-methoxycinnamyl cinnamate; (33) 9-oxo-10(E),12(Z)-octadecadienoic acid; (34) 9-oxo-10(E),12(E)-octadecadienoic acid.

**Figure 2 foods-13-03022-f002:**
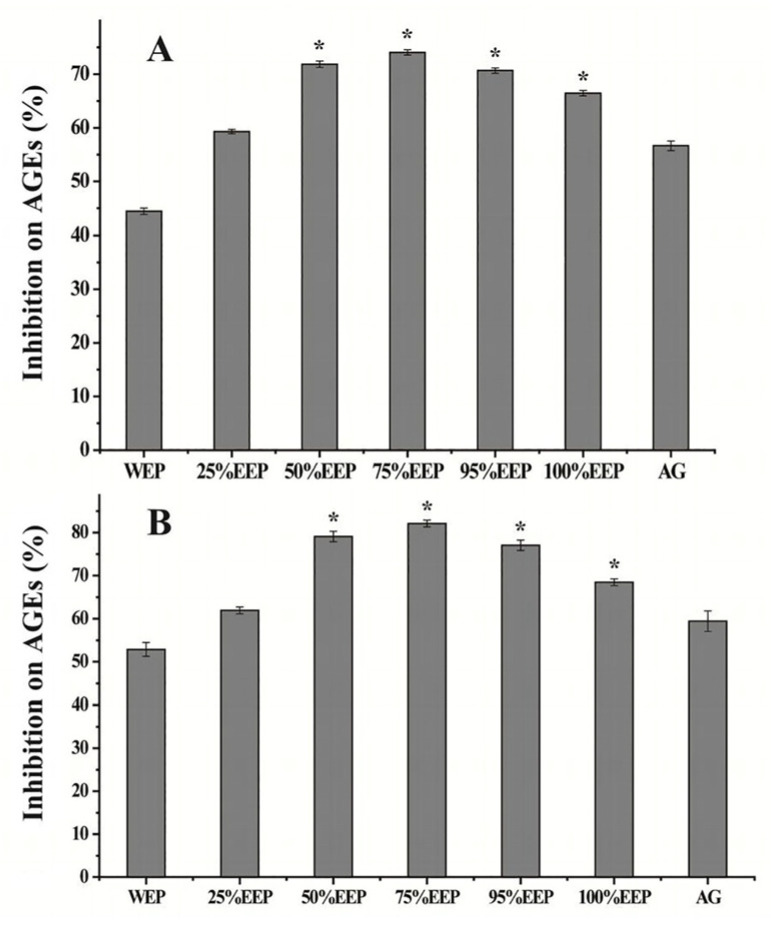
Effects of different aqueous ethanol extracts of propolis (2.5 μg/mL) or AG (1.0 mM) on the formation of total AGEs. BSA-glucose model (**A**), BSA-MGO model (**B**). * *p* < 0.05 versus positive control AG.

**Figure 3 foods-13-03022-f003:**
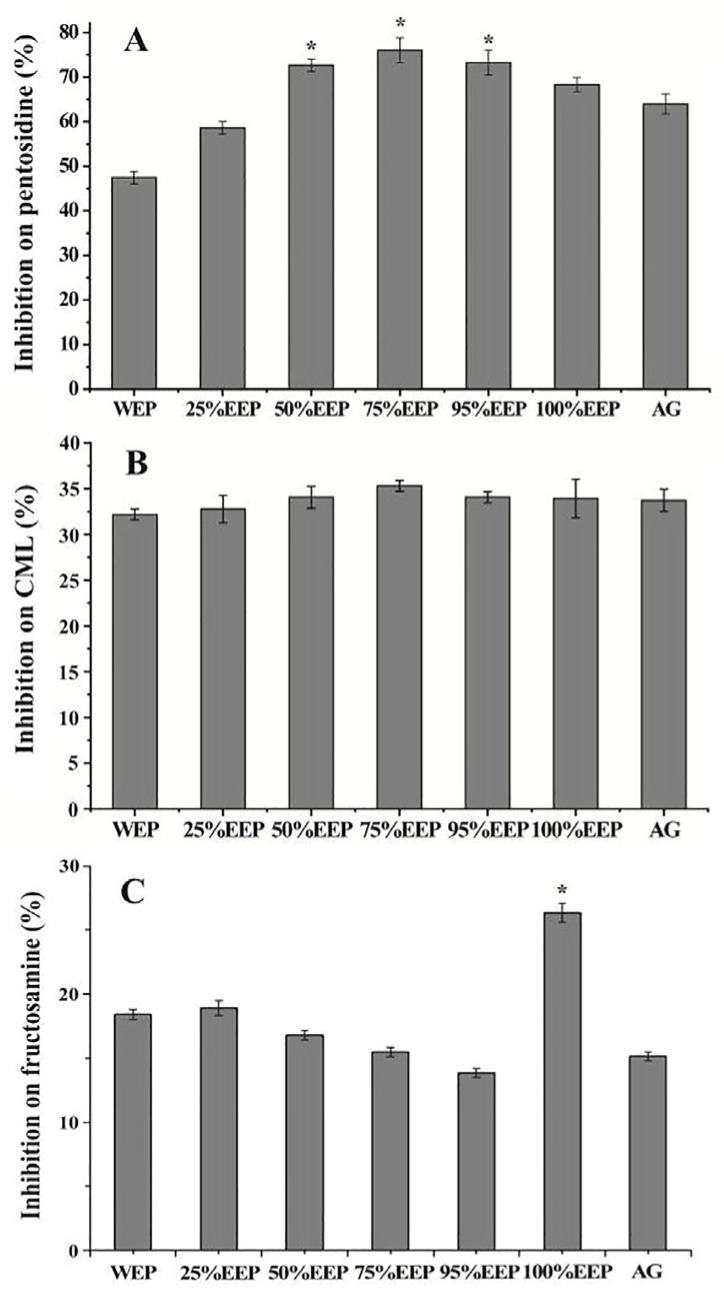
Effects of different aqueous ethanol extracts of propolis (2.5 μg/mL) or AG (1.0 mM) on the formation of pentosidine (**A**), CML (**B**) and fructosamine (**C**) in BSA-glucose model. * *p* < 0.05 versus positive control AG.

**Figure 4 foods-13-03022-f004:**
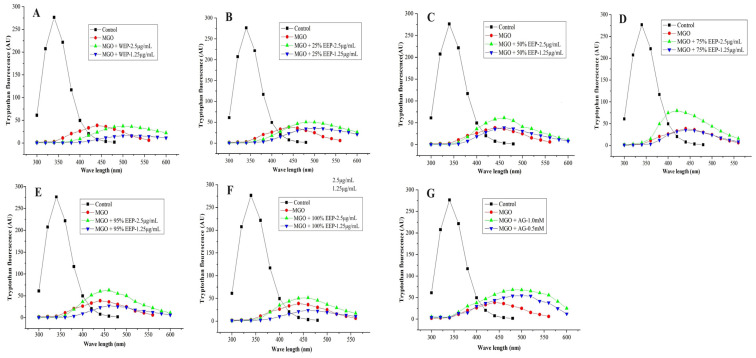
Quenching of tryptophan fluorescent spectra of BSA induced by MGO in the absence (control) and presence of different aqueous ethanol extracts of propolis (1.25, 2.5 μg/mL) or AG (0.5 mM, 1.0 mM). WEP (**A**), 25% EEP (**B**), 50% EEP (**C**), 75% EEP (**D**), 95% EEP (**E**), 100% EEP (**F**), AG (**G**).

**Figure 5 foods-13-03022-f005:**
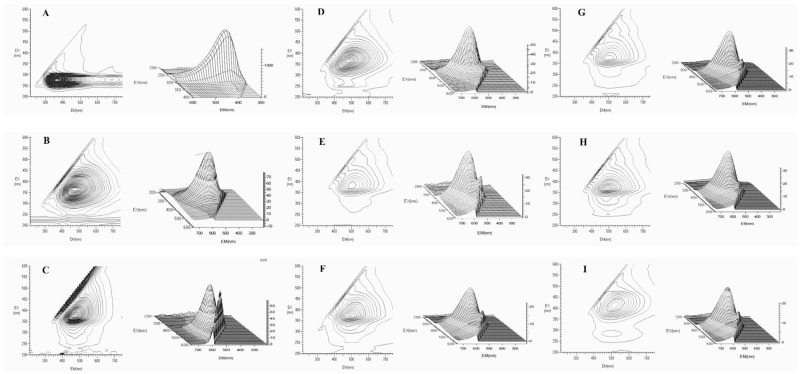
The three-dimensional fluorescence spectra and contour maps of the interaction between BSA induced by MGO in the absence and presence of different aqueous ethanol extracts of propolis (2.5 μg/mL) or AG (1.0 mM). BSA (**A**), BSA induced by MGO (**B**), BSA induced by MGO in the presence of WEP (**C**), 25% EEP (**D**), 50% EEP (**E**), 75% EEP (**F**), 95% EEP (**G**), 100% EEP (**H**) and AG (**I**), respectively.

**Table 1 foods-13-03022-t001:** Phenolics profiles of 75 wt.% ethanol extracts of propolis (mg/g dried extracts).

Peak Number	Components	Content	Peak Number	Components	Content
1	3,4-Dihydroxybenzaldehyde	0.24 ± 0.05	18	Isorhamnetin	16.25 ± 1.09
2	Caffeic acid	8.48 ± 0.42	19	Pinocembrin	106.18 ± 7.58
3	Vanillin	0.07 ± 0.01	20	Benzyl caffeate	30.72 ± 3.66
4	*p*-Coumaric acid	5.38 ± 0.36	21	Pinobanksin-3-O-acetate	196.91 ± 10.48
5	Ferulic acid	1.24 ± 0.11	22	Chrysin	62.35 ± 7.98
6	Isoferulic acid	5.86 ± 0.44	23	Phenethyl caffeate	36.61 ± 3.04
7	Benzoic acid	0.05 ± 0.01	24	Galangin	40.32 ± 4.18
8	3,4-Dimethoxycinnamic acid	3.08 ± 0.19	25	Benzyl *p*-coumarate	10.64 ± 1.05
9	Cinnamic acid	0.23 ± 0.04	26	Benzyl ferulate	1.35 ± 0.09
10	4-Methoxy cinnamic acid	2.74 ± 0.04	27	Cinnamyl caffeate	24.31 ± 2.69
11	5-Methoxy pinobanksin	5.67 ± 0.38	28	Pinostrobin	6.07 ± 0.62
12	Pinobanksin	30.53 ± 2.79	29	Tectochrysin	2.33 ± 0.21
13	Quercetin	2.14 ± 0.21	30	Cinnamyl-*p*-coumarate	9.72 ± 0.98
14	Alpinetin	2.94 ± 0.34	31	Cinnamyl cinnamate	0.51 ± 0.07
15	Kaempferol	0.04 ± 0.01	32	4-Methoxy cinnamyl cinnamate	0.72 ± 0.08
16	Cinnamylidene acetic acid	1.43 ± 0.11	33	9-oxo-ODE	35.67 ± 2.58
17	Apigenin	1.94 ± 0.22	34	9-oxo-ODA	5.72 ± 0.48

**Table 2 foods-13-03022-t002:** The scavenging abilities of different aqueous ethanol extracts of propolis on hydroxyl radicals generated through glucose autoxidation.

Different Aqueous Ethanol Extracts of Propolis	Scavenging of Hydroxyl Radical IC_50_ (μg/mL)	The Content Sum of 13 Flavonoids (mg/g Dried Extracts)	The Content Sum of 19 Phenolic Acids and Their Esters (mg/g Dried Extracts) (mg/g Dried Extracts)
Water extracts of propolis (WEP)	17.44 ± 0.47 ^f^	3.70 ^c^	60.26 ^c^
25% ethanol extracts of propolis (25% EEP)	14.99 ± 0.29 ^e^	63.35 ^b^	83.21 ^b^
50% ethanol extracts of propolis (50% EEP)	13.58 ± 0.20 ^d^	418.81 ^a^	116.78 ^a^
75% ethanol extracts of propolis (75% EEP)	9.99 ± 0.14 ^a^	473.67 ^a^	143.38 ^a^
95% ethanol extracts of propolis (95% EEP)	10.90 ± 0.25 ^b^	441.87 ^a^	149.67 ^a^
100% ethanol extracts of propolis (100% EEP)	12.73 ± 0.05 ^c^	400.97 ^a^	175.79 ^a^

Note: The IC_50_ values were calculated from the dose–response curve of six concentrations of each test compound in triplicate. Data are given as mean ± standard deviation (*n* = 3). Values with superscript letters ^a–f^ are significantly different (*p* < 0.05) between each group.

**Table 3 foods-13-03022-t003:** Effect of different aqueous ethanol extracts of propolis on the levels of carbonyl and thiol groups in BSA oxidative modifications.

Samples	Protein Carbonyl (nmol/mg Protein)	Thiol Group (pmol/mg Protein)
Control A	0.46 ± 0.02	10.23 ± 0.47
Control B	1.91 ± 0.04	3.86 ± 0.13
Water extracts of propolis (WEP)	1.65 ± 0.03 *	6.92 ± 0.61 *
25% ethanol extracts of propolis (25% EEP)	1.58 ± 0.03 *	6.94 ± 0.21 *
50% ethanol extracts of propolis (50% EEP)	1.37 ± 0.05 *	7.13 ± 0.05 *
75% ethanol extracts of propolis (75% EEP)	1.22 ± 0.05 *	7.98 ± 0.38 *
95% ethanol extracts of propolis (95% EEP)	1.50 ± 0.02 *	6.23 ± 0.25 *
100% ethanol extracts of propolis (100% EEP)	1.67 ± 0.01 *	6.44 ± 0.03 *
Aminoguanidine (AG)	1.71 ± 0.04 *	7.89 ± 0.18 *

Note: Results are shown as the mean ± SD (*n* = 3). The control A refers to BSA incubation without glucose; the control B refers to BSA incubation in the presence of 800 mM glucose. Significant differences (*p* < 0.01) from control B are marked with *.

## Data Availability

The original contributions presented in the study are included in the article, further inquiries can be directed to the corresponding author.

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
