# Peer review of "Inhibitory Effects of Aqueous Ethanol Extracts of Poplar-Type Propolis on Advanced Glycation End Products and Protein Oxidation"

_foods, 2024, doi:10.3390/foods13193022_

Round 1
Reviewer 1 Report
Comments and Suggestions for Authors
The possibility of using propolis extracts to reduce protein glycation seems to be a very interesting topic due to the fact that diabetes affects an increasing number of people in the world. I think that the work is well written, the methods are well described, the results are properly shown and discussed.
I have one suggestion: The authors could have emphasized more about the application of these results, how we could use them directly.
I have noticed some minor errors:
Latin names should be italicized, e.g. Apis mellifera, Populus canadensis, in vitro, etc.
The names of chemical compounds should also be written correctly: CuSO4, not CuSO4, NaN3, etc.
Line 326 – “In this study, the BAS-Glucose and BSA-MGO models were utilized to assess the anti-AGEs effects of aqueous” – “utilized? – maybe it will be better to change this word into another.
Line 403 – “demonstrat” ?
Author Response
- I have one suggestion: The authors could have emphasized more about the application of these results, how we could use them directly.
Response: According to the suggestion, we have added relevant descriptions in the revised manuscript. “Previous research has established a connection between elevated levels of AGEs in human tissues and the development of various diseases such as obesity, diabetes, inflammation, and cardiovascular conditions. This underscores the importance of AGE inhibition for managing these health issues. Due to the significant adverse effects of synthetic AGE inhibitors, there has been a growing interest in discovering and developing natural alternatives in recent years. Our current work suggests that the natural product propolis, especially poplar-type propolis, may offer a viable strategy for AGE inhibition. Given its specific phenolic compounds, poplar-type propolis has the potential to be a promising candidate as an AGE inhibitor in dietary supplements and nutraceuticals. In the future, we will focus on clinical trials to evaluate the efficacy of poplar-type propolis in inhibiting AGEs in vivo, providing reliable evidence for the development of novel propolis nutraceuticals aimed at preventing and treating chronic diseases associated with AGEs.” (lines 534-545). Thanks for the suggestion.
- I have noticed some minor errors: Latin names should be italicized, e.g. Apis mellifera, Populus canadensis, in vitro, etc.The names of chemical compounds should also be written correctly: CuSO4, not CuSO4, NaN3, etc.
Response: We apologize for the font errors, and these errors have been corrected in the revised manuscript (lines 57, 91, 114, 116, 156, 167, 171, 174, 189, 332). Thanks for the suggestion.
- Line 326-“In this study, the BAS-Glucose and BSA-MGO models were utilized to assess the anti-AGEs effects of aqueous”-“utilized?-maybe it will be better to change this word into another.
Response: According to the suggestion, we have revised the description from “were utilized to assess” to “were selected for evaluating” (lines 341-342).
- Line 403-“demonstrat” ?
Response: We have corrected “demonstrat” to “demonstrated” (line 423). Thanks for the suggestion.
Reviewer 2 Report
Comments and Suggestions for Authors
The manuscript contains basic in vitro models used to assess the anti-glycation potential and mechanism of hydro-alcoholic extracts of a polar-type Chinese propolis. The study was designed and reported well. The following are my comments:
What informed the choice of extract concentration (2.5 µg/mL) used in the BSA model of anti-glycation assay? Other studies on the anti-glycation property of propolis (https://doi.org/10.1080/13880209.2017.1340962; https://doi.org/10.1111/ijfs.14284) used extract concentrations up to 20 and 25 µg/mL.
Why was a different concentration range (6.25 - 100 µg/mL) of extracts used in the hydroxyl radical scavenging assay?
Why was a dose-dependent anti-glycation assay not done to determine the IC50 values of the propolis extracts. For consistency purpose, authors should consider conducting a dose-dependent anti-glycation assay as they did for the hydroxyl radical scavenging assay.
From a medicinal phytochemistry perspective, authors should comment on the predominant presence of some flavonoids like Pinocembrin, Pinocembrin, pinobanksin-3-o-acetate etc.
Author Response
- What informed the choice of extract concentration (5 µg/mL) used in the BSA model of anti-glycation assay? Other studies on the anti-glycation property of propolis (https://doi.org/10.1080/13880209.2017.1340962; https://doi.org/10.1111/ijfs.14284) used extract concentrations up to 20 and 25 µg/mL.
Response: According to the suggestion, we carefully reviewed the referenced literature. Numerous studies have shown that the bioactive compounds in propolis vary significantly depending on their geographical and botanical origins. The propolis selected in the first referenced literature was Brazilian green propolis, which is predominantly composed of artepillin C and terpenoids. Compared to the poplar-type propolis, it contains relatively fewer polyphenolic compounds, in both variety and quantity, particularly flavonoids. Although the Iranian propolis used in the second referenced literature is also of the poplar type, our previous research has revealed that poplar-type propolis can be categorized into several types based on different botanical origins. These types have distinct flavonoid profiles, and their antioxidant and other properties also vary (Foods 2023, 12 (12), 2304; Journal of Apicultural Research 2018, 57, 228-245). After a thorough comparison of the polyphenolic compound results in Table 1 of the literature, we found that the total polyphenolics content of the 75% ethanol extracts of propolis used in our research (411.83 mg/g) was approximately twice as high as that in the Iranian propolis (218.3 mg/g). Furthermore, this multiple relationship can also be observed in the total content of flavonoids, and phenolic acid and their esters, respectively. Notably, the pinobanksin-3-O-acetate in our 75% ethanol extracts of propolis, with a content of 196.91 mg/g, is nearly three times higher than the amount in the Iranian propolis (67.55 mg/g). Therefore, a relatively lower concentration (2.5 µg/mL) of propolis extract was used in our experimental investigation to assess its anti-glycation potential.
- Why was a different concentration range (6.25 - 100 µg/mL) of extracts used in the hydroxyl radical scavenging assay?
Response: In this study, we used the methods described in the literature to assess the effectiveness of various aqueous ethanol extracts of poplar-type propolis in scavenging hydroxyl radicals generated through glucose autoxidation (Biochemical Journal 1988, 256, 205-212). To more directly compare the inhibitory effects of different propolis extracts, we determined their hydroxyl radical scavenging capacity by calculating the IC50 values. This was done by adding a series of concentrations of propolis extracts to the reaction system. Based on preliminary experimental results, the extract concentrations were gradually reduced from 100 µg/mL to 50 µg/mL, 25 µg/mL, 12.5 µg/mL, and 6.25 µg/mL. According to the IC50 values presented in Table 1, this concentration range is deemed appropriate.
- Why was a dose-dependent anti-glycation assay not done to determine the IC50 values of the propolis extracts. For consistency purpose, authors should consider conducting a dose-dependent anti-glycation assay as they did for the hydroxyl radical scavenging assay.
Response: In this study, we established BSA-glucose and BSA-MGO models to assess the anti-glycation effects of different aqueous ethanol extracts of propolis, based on a method previously reported (Food Chemistry 2008, 106, 475-481). We evaluated the inhibition effects based on a single concentration of propolis extracts to calculate and compare the inhibition rates, without incorporating a dose-dependent anti-glycation assay through determining IC50 values. This was indeed an area for improvement in our future research. We sincerely appreciate the reviewer’s valuable suggestion.
- From a medicinal phytochemistry perspective, authors should comment on the predominant presence of some flavonoids like Pinocembrin, Pinocembrin, pinobanksin-3-o-acetate etc.
Response: Thanks for the suggestion. Flavonoids have taken considerable attention due to their potential health benefits. Previous literature has reported that flavonoid compounds such as chrysin, galangin, pinocembrin, pinobanksin and its ester derivatives exhibit strong antioxidant activities and are associated with reduced risk of various diseases, including heart disease, asthma, cancer, diabetes, and Alzheimer’s (Food Research International 2020, 130:108933). For instance, pinobanksin-3-cinnamate may provide neuroprotection through counteracting oxidative stress and has the potential to treat vascular dementia (Journal of Natural Medicines 2015,69:358-365). Pinocembrin has been shown to inhibit atherosclerosis progression by enhancing the level and function of endothelial progenitor cells (Cytotechnology 2013, 65:541-551). Our previous research demonstrated that galangin and pinocembrin from propolis can improve insulin resistance in HepG2 cells (Evidence-Based Complementary and Alternative Medicine 2018, 2018:7971842 ). Additionally, some flavonoids, such as quercetin and genistein, have been confirmed to inhibit advanced glycation end-products (AGEs) (Molecules 2015, 20:3309-3334). Therefore, we speculated that the high content of flavonoids in the poplar-type propolis, including pinocembrin, chrysin, galangin, and pinobanksin and its ester derivatives, etc, may be the primary trigger for its inhibitory effects of protein glycation. We have added the descriptions and references in the revised manuscript. Please see lines 269-281.
Reviewer 3 Report
Comments and Suggestions for Authors
Minor points: please order the chromatograms of Figure 1, WEP(a), 25 % EEP (b).....
Mayor points:
-Authors associated scavenging capacity of hydroxyl radicals with flavonoids contain, which is also correlated in figures 2 and 3, thus they include a new column in table 2 with the contain of flavonoids for each EEP % for a better discussion.
-In lines 360-361, authors mentioned that the highest pentosidine inhibition and CML was observed at 75 % of EEP, however there were not statistical differences for CML (lines 364-365), thus authors should clearly indicate that it does´t matter the EEP concentration there was not any effect of EEP on CML inhibition and explain wahy.
-In the case of figure 3C, authors associated the highest effect of EEP on fructosamine inhibition at 100 % with the contain of phenolic acids ans esters, they should include the concentration in (....) and compare this result with other reports.
Author Response
- Minor points: please order the chromatograms of Figure 1, WEP(a), 25 % EEP (b).....
Response: According to the suggestion, we have ordered and added the numbering of the chromatograms in Figure 1, and revised the relevant descriptions in the manuscript (lines 253-255, 263).
- Mayor points: Authors associated scavenging capacity of hydroxyl radicals with flavonoids contain, which is also correlated in figures 2 and 3, thus they include a new column in table 2 with the contain of flavonoids for each EEP % for a better discussion.
Response: Thanks for the suggestions. We have added new columns named “The content sum of 13 flavonoid” and “The content sum of 19 phenolic acids and their esters” in Tables 2 to better support the discussion in Figures 1-3, and revised the related descriptions “This viewpoint is further supported by the content sums of 13 flavonoids, 19 phenolic acids and their esters shown in Table 2.” in the revised manuscript (lines 317-319, 352, 430).
- In lines 360-361, authors mentioned that the highest pentosidine inhibition and CML was observed at 75 % of EEP, however there were not statistical differences for CML (lines 364-365), thus authors should clearly indicate that it does´t matter the EEP concentration there was not any effect of EEP on CML inhibition and explain why.
Response: Thanks for the suggestions. The results of Figure 3 indicated that, compared to pentosidine, different aqueous ethanol extracts of propolis exhibit relatively weak inhibitory activity against CML. But this does not mean that these propolis extracts have no inhibitory activity against CML, because they exhibited the inhibitory effects against CML similar to AG (inhibitory rate 33.72%), with inhibitory rates ranging from 32.19% to 35.3%, only without statistically significant differences. In addition, we have discussed the reasons for the relatively low inhibitory activity against CML in the original manuscript. Previous literature indicates that compared to pentosidine, CML can be synthesized through multiple pathways. In addition to being produced by glycosylated protein oxidation, it can also be generated through lipid peroxidation, formation of dicarbonyl compounds from glucose oxidation, and reactions with proteins or free lysine. Propolis extract and AG might inhibit CML formation by trapping a certain amount of dicarbonyl compounds and inhibiting the subsequent series of oxidation and cross-linking reactions between proteins and dicarbonyl compounds, leading to a relatively weaker inhibitory effect.
We have revised and added the related descriptions in section 3.4, please see lines 373-380, 385, 388-390, 393-397.
- In the case of figure 3C, authors associated the highest effect of EEP on fructosamine inhibition at 100 % with the contain of phenolic acids and esters, they should include the concentration in (....) and compare this result with other reports.
Response: According to the suggestion, we have added the content sums of 19 phenolic acids and their esters for different aqueous ethanol extracts of propolis in Tables 2, and revised the related descriptions in our manuscript (lines 425-429).
Among these propolis extracts, the 100% ethanol extract of propolis shows the highest sum content of 19 phenolic acids and esters (175.79 mg/g), with notably high levels of phenolic esters. The results in Figure 3C indicate that the propolis extracts exhibit a low inhibitory effect on Amadori products. Although the inhibition rate of the 100% ethanol extract of propolis is the highest (26.33%), it remains relatively low. It is known that certain phenolic acids, such as vanillic acid, caffeic acid, ferulic acid, isoferulic acid, and cinnamic acid, can effectively reduce the formation of AGEs. Note-worthily, phenolic esters have also been reported to exhibit an inhibitory effect on AGEs (Molecules 2019, 24: 2689). Previous studies indicate that, usually, the antiglycative action of flavonoids and phenolic acids is mainly attributed to their antioxidant properties and ability to trap dicarbonyls, but due to differences in the molecular structures of phenolic acids and flavonoids, their mechanisms for inhibiting AGEs may differ (Food Research International 2020, 130:108933). Currently, there are limited reports on the mechanisms of phenolic esters in glycation inhibiting, so we speculate that the inhibitory activity of 100% ethanol extracts of propolis on the formation of Amadori products may be due to its relatively higher content of phenolic acids and esters.
Round 2
Reviewer 2 Report
Comments and Suggestions for Authors
Authors have provided sufficient responses to my comments.
Reviewer 3 Report
Comments and Suggestions for Authors
The manuscript has the quality to be published